# Dead or Alive: Drivers of Wind Mortality Initiate Multiple Disturbance Regime in a Temperate Primeval Mountain Forest

**Ivana Vašíčková** [1,*]**, Pavel Šamonil** [1,2]**, Jakub Kašpar** [1]**, Andrea Román-Sánchez** [1]**, Tomáš Chuman** [3,4] **and Dušan Adam** [1]

[1] Department of Forest Ecology, The Silva Tarouca Research Institute, 602 00 Brno, Czech Republic; pavel.samonil@vukoz.cz (P.Š); jakub.kaspar@vukoz.cz (J.K.); o92rosaa@uco.es (A.R.-S.); dusan.adam@vukoz.cz (D.A.)

[2] Faculty of Forestry and Wood Technology, Mendel University, 613 00 Brno, Czech Republic

[3] Czech Geological Survey, 152 00 Prague, Czech Republic; tomas.chuman@email.cz

[4] Department of Physical Geography and Geoecology, Faculty of Science, Charles University, 128 43 Prague, Czech Republic

\* Correspondence: ivana.vasickova@vukoz.cz; Tel.: +420-737-765-250

**Abstract:** The driving forces of tree mortality following wind disturbances of mountain mixed European temperate forests belongs among issues not comprehensively resolved. Hence, we aimed to elucidate the key factors of tree resistance to historical severe disturbance events in the Boubínský Primeval Forest, one of the oldest forest reserves in the Czech Republic. By using spatially explicit tree census, dendrochronological and soil data, we study spatial and temporal patterns of past disturbances and mathematically compared selected characteristics of neighboring trees that were killed by a severe storm in 2017 and those that remained undisturbed. The tendency of trees toward falling was primarily driven edaphically, limiting severe events non-randomly to previously disturbed sites occupied by hydromorphic soils and promoting the existence of two spatially-separated disturbance regimes. While disturbed trees usually recruited in gaps and experienced only one severe release event, surviving trees characteristically regenerated under the canopy and were repeatedly released. Despite the fact that disturbed trees tended to reach both lower ages and dimensions than survivors, they experienced significantly higher growth rates. Our study indicates that slow growth with several suppression periods emerged as the most effective tree strategy for withstanding severe windstorms, dying of senescence in overaged life stage. Despite the selective impact of the Herwart storm on conifer population, we did not find any difference in species sensitivity for most characteristics studied. We conclude that the presence of such ancient, high-density wood trees contributes significantly to the resistance of an entire stand to severe storms.

**Keywords:** dendrochronology; disturbance history; *Fagus sylvatica*; *Picea abies*; geostatistics; growth release; soil properties; wind mortality

## 1. Introduction

Natural disturbances are widely acknowledged to be a primary force in the dynamics of diverse primeval forest ecosystems, shaping the forest structure and composition and maintaining species and soil diversity [1–6]. Whereas fine-scale disturbances dominate the natural forest dynamics of deciduous upland forests of Central Europe [7–10], several studies have highlighted the importance of large-scale, stand-replacing events for conifer mountain forest ecosystems [11–15]. More recently, a mixed-severity disturbance regime, predominantly driven by gap dynamics with infrequent severe stand-replacing events, has been documented for some mixed mountain forest ecosystems [8,13,16–20]. Though fine-scale processes have been abundantly studied in Central European forests, the long-

term dynamics of mountain sites following high-severity disturbances needs further investigation, particularly in respect to tree-species coexistence, spatial patterns, tree-soil interactions and factors controlling mortality. In addition, water-affected sites have been largely overlooked in analyses for most of the above cited studies, preventing comprehensive insight into large-scale disturbances.

Disturbance regimes of natural forest ecosystems depend on a number of factors that determine the predisposition of trees to biotic or abiotic disturbances. It is widely acknowledged that larger and older trees are more predisposed to disturbance (e.g., [21–23]). Another element predicting the stand resistance to wind mortality is species composition [24,25], primarily through differences in shape of root system of various tree species [26], as well as their level of shade tolerance [27]. Based on trade-offs between growth rates and tree lifespan observed by some authors [28–31], we assume preceding radial growth as one of the key factors of tree stability. Nevertheless, as far as we know, no previous research has aimed at studying tree-ring patterns and disturbance histories to describe wind mortality following extreme windstorms in any type of forest ecosystems. Pavlin et al. [32] quantified the relationship between disturbance history and tree-growth patterns and tree lifespan across primeval forests of Carpathian Mountains, but they did not provide clear evidence that prolonged periods of suppression of tree in understory contributes to its greater mechanical resistance during the strong winds.

Soil plays a crucial role in the resistance of trees to uprooting by providing anchorage to the tree. Water-saturated soils are regarded as giving less support to trees than well drained soil units [33,34], primarily due to limited rooting depth [35]. Thus, some forest sites are potentially more susceptible to wind mortality than others [3,36], promoting the pattern of repeatedly disturbed soils. This preference results in a non-random spatial distribution of single tree throw events markedly contributing to the structural complexity of primeval forests. Recent studies have revealed exceptional spatial pedocomplexity in some mountain old-growth temperate forests. For instance, Daněk et al. [37] found 37 soil units and exceptional spatial soil pedocomplexity within the Boubínský Primeval Forest (hereinafter Boubin). Similarly, Šamonil et al. [38] and Valtera et al. [39] found extremely short ranges of soil spatial autocorrelation in beech- and spruce-dominated primeval forests. The sources of such high soil variability and diversity and their roles in forest dynamics are unclear [40] and cannot be comprehensively explicated by conventional pedogenetic factors. Šamonil et al. [41] proposed that the effect of individual trees triggers such heterogeneity, promoting the occurrence of several disturbance regimes covering different soil units within one forest.

Historical documents on severe windstorms [42–46] demonstrate that Czech highlands and mountains have suffered heavy windstorms at least once per century, profoundly affecting many forests and sometimes reinforced by subsequent biotic disturbances such as spruce bark-beetle outbreaks. By the volume of damaged wood, the four most important stand-replacing events of last centuries in the Czech Lands are windstorms in 1740, 1833, 1868 and 1870. The Kyrill (2007), Emma (2008) and Herwart (2017) storms provide more recent examples. With the exception of the Kyrill storm, all the above-mentioned events left noticeable footprints on the structure of the mixed mountain forest at Boubin, one of the oldest forest reserves in Europe, under strict protection since 1858. As far as we know, the forest has never been cut or otherwise directly affected by humans.

In October 2017, the Boubin Reserve was hit by an extreme severe windstorm named Herwart, affecting the forest structure with hectare-sized blowdowns, as well as many small canopy gaps. Still, an undisturbed closed-canopy forest was preserved on a majority of the original area. This selective effect of the storm raised the question why some trees are more susceptible to toppling than others. We therefore attempted to examine the crucial agents of mortality and the survival dynamics of severe windstorms in Boubin, comparing the properties of habitats and trees that were killed by Herwart (hereafter termed "disturbed") and those that remained intact after the windstorm (hereafter "survivors").

We provide a comprehensive study considering most important mortality factors cited before in literature, supplemented with disturbance history of trees and their preceding radial growth that is lacking in Central European region. Specifically, we formulated the following hypotheses:

**Hypothesis 1.** (**H1**): *Site hypothesis. Evidence: severe disturbances are significantly associated with the areas occupied by (semi-)hydromorphic soil.*

**Hypothesis 2.** (**H2**): *Historical hypothesis. Evidence: disturbed trees experience different disturbance histories compared to survivors.*

**Hypothesis 3.** (**H3**): *Species hypothesis. Evidence: some tree species are disproportionally disturbed by a strong storm.*

**Hypothesis 4.** (**H4**): *Size hypothesis. Evidence: wind primarily affects larger and older tree individuals.*

**Hypothesis 5.** (**H5**): *Material hypothesis. Evidence: smaller growth increments contribute to tree resistance to strong winds.*

## 2. Materials and Methods

### 2.1. Study Site

Our study took place in Boubin, one of the largest forest reserves of the Czech Republic. The core zone of the reserve comprising 46.62 hectares is the best example of a Central European primeval mountain forest without any past direct human interventions [47]. Boubin is located in the Šumava mountain range on the crystalline Bohemian Massif in the southern part of the country (Figure 1), formed by migmatite. It is situated on the north-eastern slopes of Boubin Mountain (1362 m a.s.l.), spanning from 925 to 1108 m a.s.l. Mean annual precipitation is 1067 mm and mean temperature is 4.9 °C. Cambisols and Podzols are the most frequent soil units in terrestrial areas, whereas Gleysols, Stagnosols and Histosols occupy spring areas saturated by water. Since the early Holocene, the forests had been dominated by *Picea abies* (L.) Karsten [48]; however, over the last centuries an accelerated increase in the proportion of *Fagus sylvatica* L. has occurred [49]. The tree layer is currently dominated by *Fagus sylvatica* (61.6% of tree individuals), followed by *Picea abies* (36.1%) and *Abies alba* Mill. (1.7%) [50]. Plant communities can be classified as *Calamagrosio-villosae Fagetum* and *Calamagrostio villosae Piceetum* [51].

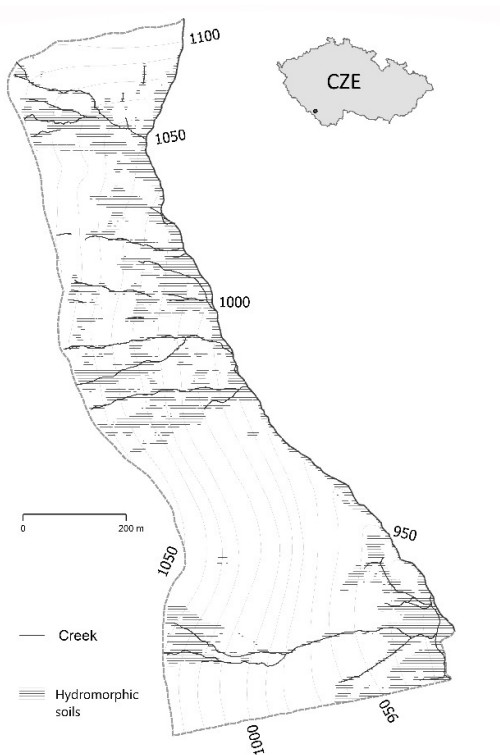

**Figure 1.** Map of the Boubin old-growth forest and location within the Czech Republic.

*2.2. General Conceptual Model*

To elucidate factors of mortality in Boubin, we posed five hypotheses, spatially structured from stand- ($H_1$) to gap- ($H_2$–$H_3$) and, finally, tree-level ($H_3$–$H_5$) (Figure 2). As it is not possible to strictly separate the mutual effect of individual factors, these formulated hypotheses may intersect and cannot be necessarily reciprocally excluded. While $H_1$ was focused on whole-stand data, for verification of $H_2$–$H_5$ we chose neighboring pairs of cored disturbed and survivor trees at a maximum distance of 15 m from each other to detect mortality factors at the finest scale. The analyses and data sets used for particular hypotheses are shown in Table 1.

**Table 1.** A list of hypotheses, including analyses and data sets used.

| Hypothesis | H1 Site | H2 Historical | H3 Species | H4 Size | H5 Material |
|---|---|---|---|---|---|
| Spatial effect | Stand-level | Gap-level | Gap-/Tree-level | Tree-level | Tree-level |
| Approach | Dendrochronology Soil science Geostatistics | Dendrochronology | Tree census | Tree census Dendrochronology | Dendrochronology |
| Sample size N | 1061 cores 954 soil profiles | 271 pairs (release) 221 pairs (gap origin) | 271 pairs | 271 pairs (DBH) 221 pairs (age) | 271 pairs |
| Statistical test | Mann-Whitney | Paired *t*-test (release) McNemar's test (gap origin) | Chi-sqare test | Paired *t*-test | Paired *t*-test |

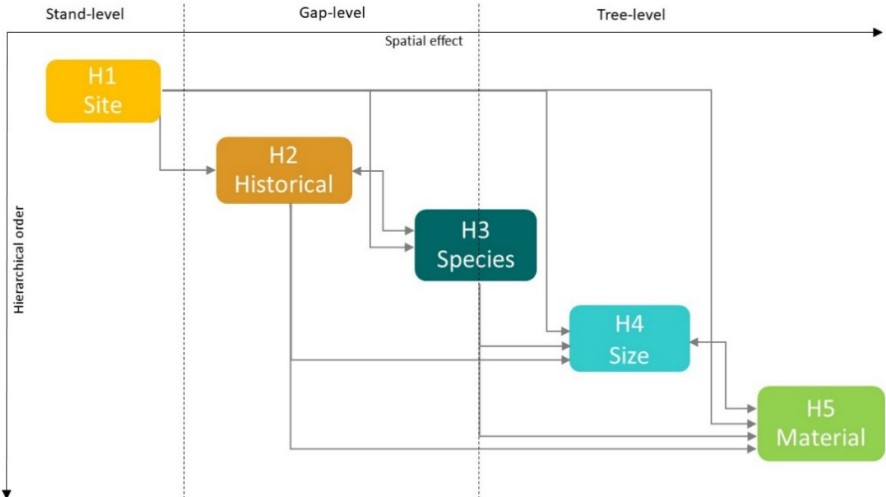

**Figure 2.** General conceptual model including relations between individual hypotheses.

*2.3. Sampling Design and Data Processing*

A detailed tree census, carried out in 2019 within the core area of 46.62 hectares, served as the basis for the dendrochronological survey as well as geostatistical analysis ($H_1$). The locations of circa 17.000 all living and dead trees with diameter at breast height ≥10 cm including specific tree properties (dimension, species and health status) provide a unique data set to study forest dynamics.

The soil survey verifying $H_1$, established in a grid in 2015 [37], provided detailed description and classification of 954 soil profiles according to the World Reference Base [52]. Soil hydromorphism was expressed by the presence of stagnic, gleyic and histic soil properties and in some cases even diagnostic soil horizons. In total, 11 degrees of soil hydromorphism were recognized in Boubin [37]. Terrestrial soils were defined as from the 0 to 3rd degrees of the water gradient (Cambisols, Podzols), hydromorphic soils were determined as from the 4th to 11th degree (Stagnosols, Gleysols, Histosols).

To study differences in the disturbance history of trees along the gradient of soil hydromorphism ($H_1$), as well as disturbed and survivor trees ($H_2$), we used dendrochronological data collected for research into the Boubin disturbance history [20]. First, we employed a collection of 400 core series extracted on a regular grid in 2012 from living canopy trees or trees exposed in gaps. This data set served as a basis for the evaluation of survivor trees. Since 2013 and in particular after the Herwart storm, recently disturbed trees could be cored with no limitation ,and thus, we sampled an extensive set of 1204 increment cores from 717 trees, serving for assessment of disturbed trees. Concerning canopy position, we assumed that these trees were predominantly exposed, as derived from the relationship of dimension and canopy position developed by Kašpar et al. [20] for the whole Boubin area. For both datasets, we used one core per tree at height 0.5–1.3 m. Only series without mechanical damage were accepted for further analysis, resulting in 1061 increment cores. These cores were used within $H_1$ and to verify $H_2$ we employed 304 trees forming 271 pairs. As additional supporting evidence of past disturbance events, we additionally sampled 71 juvenile trees of *F. sylvatica* and 45 juvenile individuals of *P. abies* for calculations of the threshold for gap origin [20].

In the laboratory, increment cores were dried and sanded to improve the visual quality of the wood structure for measuring procedures. Whereas tree-ring widths of cores sampled in the period 2012–2015 were recorded using the PAST 4 software (SCIEM, 2007, Austria), core series extracted later were scanned using an Epson LA2400 scanner at 1600 DPI resolution and consequently measured in WinDENDRO software (Regent Instruments, 2014, Quebec, Canada). In both cases, tree-ring widths were registered with 0.01

mm accuracy. Following measurement, core series were cross-dated using the marker year method [53], with confirmation in COFECHA software [54].

### 2.4. Site Hypothesis (H$_1$)

We expected that wind mortality would be driven by soil water content expressed by soil units. To verify this hypothesis, we classified each cored tree ($n$ = 1061) into soil hydromorphism classes (Section 2.3) and tested if trees disturbed by the Herwart storm occupied wetter sites with comparison to survivor trees. Subsequently, to integrate one singular event into the whole disturbance history of the forest, we studied if different disturbance regimes exist under contrasting soil moisture conditions, primarily from the point of view of their time and spatial pattern.

To uncover specific disturbance regimes under different soil moisture conditions, we elaborated decadal summary disturbance histories separately for terrestrial and hydromorphic soils. For every cored tree, two types of disturbance legacies in tree-ring sequences were evaluated: (i) juvenile tree growth in a gap or under the canopy, i.e., gap origin; and (ii) reaction of the tree to the death of a nearby overtopping tree during subsequent radial growth in the form of abrupt growth changes, i.e., release.

The threshold for gap origin was calculated as in Šamonil et al. [16] according to the standard approach by Lorimer et al. [55]. The threshold for *P. abies* and *A. alba* above which trees were assumed to have been established in gap was 1.60 mm, for *F. sylvatica* 1.53 mm. Only increment cores with ≤3 cm deviation to the pith were included to the analysis of juvenile growth.

For the detection of pulses in radial growth, i.e., releases, to a nearby disturbance event, we used a boundary line approach (BL, [56]). This method, based on a comparison of the percent growth change (GC) between two 10-year intervals [57] with the average prior radial growth over the past 10 years (PG), accounts for individual factors of every sampled tree as species, social status, age and dimension. For more information on calculation of boundary line see Methods S1. Based on intensity we differentiated releases as weak (BL ≥ 25%), moderate (BL ≥ 50%) and, finally, major (BL ≥ 100%). We accepted all the releases within the examined period, i.e., repeated releases were not rejected [16,58]. As a measure of the severity of past disturbances, the proportion of canopy area disturbed (CAD, m$^2$) in a particular decade was applied. For that purpose, we first modelled the relation between diameter at breast height and current exposed crown area separately for every tree species [20]. Based on the knowledge of the canopy area of a tree in the year of disturbance and sum of exposed canopy area of all trees (sample depth), we estimated the proportion of disturbed canopy area separately for different release intensity. Finally, a graph of summary disturbance history was created separately for hydromorphic and terrestrial soils. For each decade, proportion of canopy area disturbed (upper part) was linked to juvenile growth (bottom part), providing a complex picture on past disturbance dynamics. As we attempted to eliminate false lateral releases of exposed canopy trees [58], the evaluation of disturbance history was strictly limited to tree-ring segments that had not reached the canopy. Following this restriction, we calculated the threshold of diameter, beyond which the probability was less than 5% that a tree could have been overtopped before the growth release, according to the method of Lorimer and Frelich [59]. The resulting DBHs were 55 cm for *F. sylvatica* and 56 cm for *P. abies* and *A. alba* [20]. Thus, tree-ring sequences exceeding these thresholds were excluded from further analysis.

To reveal differences in the spatial pattern of disturbance events on terrestrial and hydromorphic soils, we used a geostatistical approach. The precise location of hundreds of cored trees of known disturbance histories allowed us to study the spatial autocorrelation of a severe disturbance. For this analysis, we chose the decades 1870–1889 as the period with the parallel impact of several severe disturbances (strong windstorms, bark beetle). In contrast, the more recent Herwart storm in 2017 was not analyzed, as an insufficient time interval had passed for the full development of mortality processes. Within this

time frame, we computed variograms separately for (i) weak releases and (ii) moderate to major releases. For more information on variograms calculation, see Methods S2.

*2.5. Historical Hypothesis (H$_2$)*

We hypothesized that trees killed by the Herwart storm expressed a specific disturbance history in comparison with trees that remained intact, specifically that the frequency and severity of windstorms, which every tree records in the form of tree-ring features, would be different. This supplements the site hypothesis to a certain extent. However, unlike H$_1$ that focuses on whole-stand data, H$_2$ gives us detailed information on past disturbance dynamics at the local scale, accounting for biogeomorphological particularities. Because every disturbance is detectable only to a limited distance from the disturbed tree, we chose neighboring pairs of cored disturbed and survivor trees at a maximum distance of 15 m from each other for testing ($n$ = 271 pairs in the case of release, $n$ = 221 pairs in the case of gap origin).

We evaluated the frequency of disturbances as the number of reactions within the examined period, differentiated by weak (BL ≥ 25%) and moderate to major releases (BL ≥ 50%). As a measure of the severity of a past disturbance event, we computed the gap origin, total sum of boundary line (%) and canopy area disturbed (m$^2$) in the year of each release. A list of all evaluated variables including their basic descriptive characteristics is shown in Table 2. To ensure comparability between disturbed and undisturbed trees within the pair, the calculation of disturbance history was restricted entirely to tree-ring sequences covering the same time interval, i.e., only mutually overlapping segments were examined.

**Table 2.** Results of paired tests for Historical, Size and Material hypotheses, including the descriptive statistics for all examined variables and type of paired test used. Rows contain descriptive statistics for disturbed and survivor datasets and their differences, as demonstrated by median, 95% confidence interval width and *p*-value of paired tests (right side of the table). Tests were performed for all tree species together as well as separately for particular species pairs.

| Variable. | Min | Max | Mean | Median | SD | 95% CI | Type of Paired Test | Sample Size | All Species | Spruce-Spruce | Spruce-Beech | Beech-Spruce | Beech-Beech |
|---|---|---|---|---|---|---|---|---|---|---|---|---|---|
| No. of releases ≥25% boundary line per tree | | | | | | | | | | | | | |
| Disturbed | 0 | 5 | 0.85 | 1 | 0.96 | 0.73–0.96 | - | 271 | - | - | - | - | - |
| Survivors | 0 | 7 | 1.20 | 1 | 1.18 | 1.06–1.34 | - | 271 | - | - | - | - | - |
| Difference | - | - | - | −0.50 | - | −0.99–−<0.001 | Wilcoxon test | 542 | <0.001 | 0.005 | <0.001 | 0.523 | 0.086 |
| No. of releases ≥ 50% boundary line per tree | | | | | | | | | | | | | |
| Disturbed | 0 | 2 | 0.19 | 0 | 0.44 | 0.14–0.24 | - | 271 | - | - | - | - | - |
| Survivors | 0 | 3 | 0.37 | 0 | 0.67 | 0.29–0.45 | - | 271 | - | - | - | - | - |
| Difference | - | - | - | −0.50 | - | −0.99–−<0.001 | Wilcoxon test | 542 | <0.001 | <0.001 | 0.012 | 0.233 | 0.280 |
| Total sum of boundary line (%) per tree | | | | | | | | | | | | | |
| Disturbed | 0 | 210.93 | 32.94 | 26.71 | 40.32 | 28.12–37.76 | - | 271 | - | - | - | - | - |
| Survivors | 0 | 274.11 | 49.78 | 37.64 | 54.60 | 43.25–56.31 | - | 271 | - | - | - | - | - |
| Difference | - | - | - | −22.14 | - | −30.63–−12.41 | Wilcoxon test | 542 | <0.001 | <0.001 | <0.001 | 0.293 | 0.108 |
| Total sum of canopy area disturbed (m²) | | | | | | | | | | | | | |
| Disturbed | 0 | 108.42 | 12.34 | 1.76 | 19.71 | 9.98–14.70 | - | 271 | - | - | - | - | - |
| Survivors | 0 | 253.61 | 30.49 | 13.18 | 43.20 | 25.32–35.66 | - | 271 | - | - | - | - | - |
| Difference | - | - | - | −17.66 | - | −23.61–−12.14 | Wilcoxon test | 542 | <0.001 | 0.002 | <0.001 | 0.148 | 0.030 |
| Gap origin | | | | | | | | | | | | | |
| Disturbed | - | - | - | - | - | - | - | 221 | - | - | - | - | - |
| Survivors | - | - | - | - | - | - | - | 221 | - | - | - | - | - |
| Difference | - | - | - | - | - | - | McNemar's test | 442 | <0.001 | 0.091 | <0.001 | 0.617 | 1 |
| Diameter at breast height (cm) | | | | | | | | | | | | | |
| Disturbed | 10 | 110 | 51.99 | 52 | 23.90 | 49.13–54.84 | - | 271 | - | - | - | - | - |
| Survivors | 10 | 129 | 61.75 | 58 | 23.55 | 58.94–64.57 | - | 271 | - | - | - | - | - |
| Difference | - | - | - | −9.50 | - | −13.50–−5.50 | Wilcoxon test/*t*-test | 542 | <0.001/- | <0.001/- | 0.048/- | -/<0.001 | -/0.436 |
| Age at breast height (yrs) | | | | | | | | | | | | | |
| Disturbed | 32 | 567 | 201 | 178 | 84.94 | 190.17–212.69 | - | 221 | - | - | - | - | - |
| Survivors | 34 | 428 | 228 | 235 | 81.02 | 216.97–238.45 | - | 221 | - | - | - | - | - |
| Difference | - | - | - | −30.50 | - | −43.00–−17.50 | Wilcoxon test/t-test | 442 | <0.001/- | 0.012/- | -/<0.001 | -/0.070 | -/1 |
| Tree-ring width (RWI) | | | | | | | | | | | | | |
| Disturbed | 0.51 | 1.09 | 0.93 | 0.95 | 0.09 | 0.92–0.94 | - | 271 | - | - | - | - | - |
| Survivors | 0.58 | 1.05 | 0.87 | 0.89 | 0.10 | 0.86–0.89 | - | 271 | - | - | - | - | - |
| Difference | - | - | - | 0.05 | - | 0.04–0.07 | Wilcoxon test/*t*-test | 542 | <0.001/- | <0.001/- | <0.001/- | -/0.483 | -/0.030 |

## 2.6. Species Hypothesis (H₃)

To evaluate the role of tree species in the tendency to experience treefall, we compared the distribution of woody species between the disturbed and survivor tree dataset that represents the whole-stand tree species structure. A Pearson's Chi-square test was used to assess differences between actual disturbances of tree species populations by wind and expectations based on their proportions in the community. The null hypothesis assumes no differences between the structure of the forest community and the structure of killed trees by a storm. Testing criterion for accepting the null hypothesis was $p > 0.05$.

In addition, in order to express the species-specific impact on drivers of wind mortality, we tested if disturbance history and size, age and tree-ring width differ between disturbed and survivor trees. Thus, all the pair tests were performed separately for each combination of tree species within the pair: (i) spruce disturbed vs. spruce survivor ($n$ = 137, hereafter spruce-spruce), (ii) spruce disturbed vs. beech survivor ($n$ = 84, hereafter spruce-beech), (iii) beech disturbed vs. spruce survivor ($n$ = 17, hereafter beech-spruce) and (iv) disturbed beech vs. survivor beech ($n$ = 14, hereafter beech-beech). Fir was not evaluated as there were an inadequate number of individuals for proper statistical analyses.

## 2.7. Size Hypothesis (H₄)

Under this hypothesis, we attempted to reveal the mortality differences for neighboring pairs through tree-growth characteristics, i.e., diameter at breast height (DBH, cm) and age at breast height (yrs). While DBH was derived from whole tree census, age was estimated on the basis of measured tree-ring sequences including the number of rings to the centre for cores that missed the pith. The number of rings was calculated using the pith locator [60]. Only increment cores with deviation <3 cm to the pith were accepted for age analysis ($n$ = 221). The number of rings at 1.3 m was referred as recruitment age based on standard dendroecological approaches [57,59]; thus, former growth was not analyzed. According to Šamonil et al. [16] average age a tree reached at breast height is 8 years for *Fagus sylvatica* and 11 years for *Picea abies*.

## 2.8. Material Hypothesis (H₅)

With this hypothesis, we assessed if past growth rate enhances a tree's susceptibility to fall. We choose mean tree-ring width as the appropriate characteristic to express mechanical wood properties, e.g., wood density and strength [61]. While measurements of maximum latewood density (MXD), X-ray densitometry [62] or blue intensity [63] may be superior proxies of wood density, these methods were primarily developed for dendroclimatic purposes and are as rather costly and time consuming and beyond the scope of this study.

Every increment core within the pair ($n$ = 271) was first divested of any age-related trends by fitting a negative exponential curve to raw ring-width measurements [64]. By this procedure, tree-ring chronologies were transformed to dimensionless tree-ring width indices (RWI) and averaged. Standardization was performed using the dplR package [65] in R, ver. 4.0.4 (R Development Core Team, 2019). Average RWI values of disturbed trees were then pairwise compared with survivor individuals.

## 2.9. Statistical Analyses

The statistically significant differences in soil hydromorphism between disturbed and survivor trees (H₁) were assessed using the non-parametric Mann-Whitney test. To pairwise verify H₂–H₅, we used paired t-tests (Table 2). First, normality was checked by the Shapiro–Wilk test, and if the studied parameter did not follow a normal distribution, we applied the non-parametric paired Wilcoxon test. For gap origin calculations, a two-

dimensional contingency table was built to perform McNemar's chi-squared test. All statistical analyses were conducted using R software, ver. 4.0.4 (R Development Core Team, 2019) and determined as statistically significant if $p < 0.05$.

## 3. Results

### 3.1. Site Hypothesis ($H_1$)

We validated the site hypothesis ($H_1$) that severe disturbances most likely occur in areas occupied by (semi-)hydromorphic soils. Trees killed by the Herwart storm occupied significantly wetter sites compared to survivor trees ($p < 0.001$), with the mean degree of soil hydromorphism being 3.78 for the positions of disturbed trees and 2.74 for survivors. Similarly, 45.7% of killed trees occupied class 1 with the terrestrial relatively dry soils, but up to 65% survivor trees were found at such sites (Figure 3). On the contrary, 29% of disturbed trees grew in the most water-saturated areas (classes 8–11, i.e., Gleysols and Histosols), whereas only 18% of survivor trees appeared at those sites.

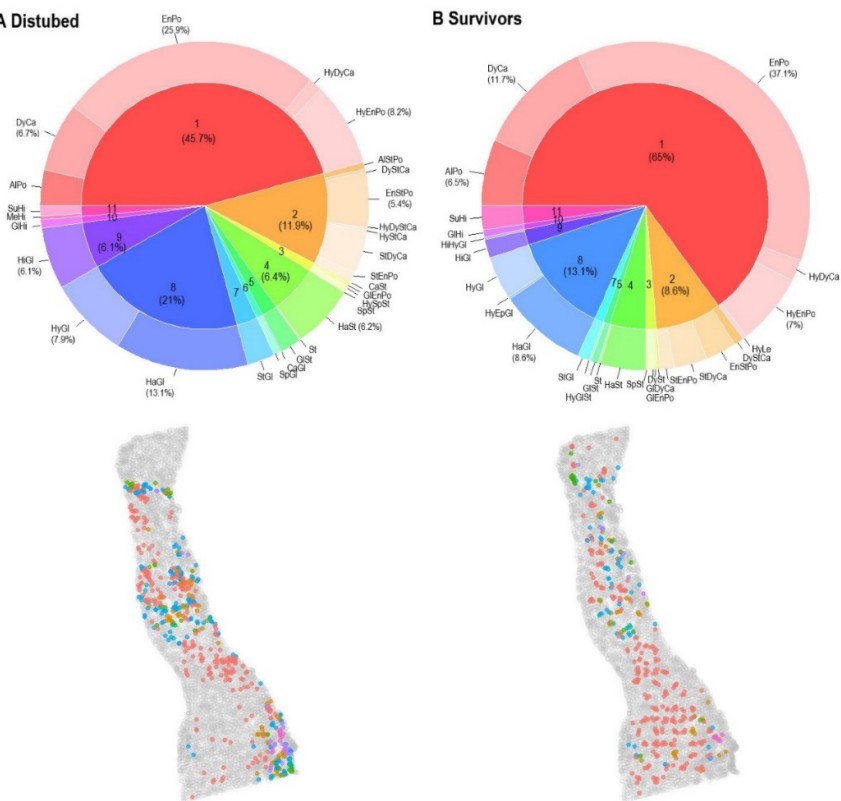

**Figure 3.** Pie and donut charts of the proportions of soil hydromorphism classes (inner pie) and soil units (outer donut) with maps of their distribution in the Boubin old-growth forest, classified by: (**A**) disturbed and (**B**) survivor trees. Proportions are depicted only if the percentage ≥5%. AlPo = Albic Podzols, DyCa = Dystric Cambisols, EnPo = Entic Podzols, HyDyCa = Hyperskeletic Dystric Cambisols, HyEnPo = Hyperskeletic Entic Podzols, HyLe = Hyperskeletic Leptosols, AlStPo = Albic Stagnic Podzols, DyStCa = Dystric Stagnic Cambisols, EnStPo = Entic Stagnic Podzols, HyDyStCa = Hyperskeletic Dystric Stagnic Cambisols, HyStCa = Hyperskeletic Stagnic Cambisols, StDyCa = Stagnic Dystric Cambisols, StEnPo = Stagnic Entic Podzols, CaSt = Cambic Stagnosols, DySt = Dystric Stagnosols, GlDyCa = Gleyic Dystric Cambisols, GlEnPo = Gleyic Entic Podzols, HySpSt = Hyperkeletic Spodic Stagnosols, SpSt = Spodic Stagnosols, HaSt = Haplic Stagnosols, St = Stagnosols, GlSt = Gleyic Stagnosols, HyGlSt = Hyperskeletic Gleyic Stagnosols, CaGl = Cambic Gleysols, SpGl = Spodic Gleysols, StGl = Stagnic Gleysols, HaGl = Haplic Gleysols, HyEpGl = Hyperskeletic Epileptic Gleysols, HyGl = Hyperskeletic Gleysols, HiGl = Histic Gleysols, HiHyGl = Histic Hyperskeletic Gleysols, GlHi = Gleyic Histosols, MeHI = Mezic Histosols, SuHI = Supric Histosols.

Maps of the spatial distribution of severe disturbances (Figure 4A,B) indicate a similarity in the spatial pattern of the Herwart windstorm in 2017 with series of disturbances in the 1870s. In addition, both events were apparently concentrated in areas occupied by (semi-)hydromorphic soils. This finding suggests that these severe events did not affect the forest randomly, but instead hit the same places repeatedly, i.e., patches with a predominance of hydromorphic soils, suggesting a tight soil-disturbance dynamics interaction.

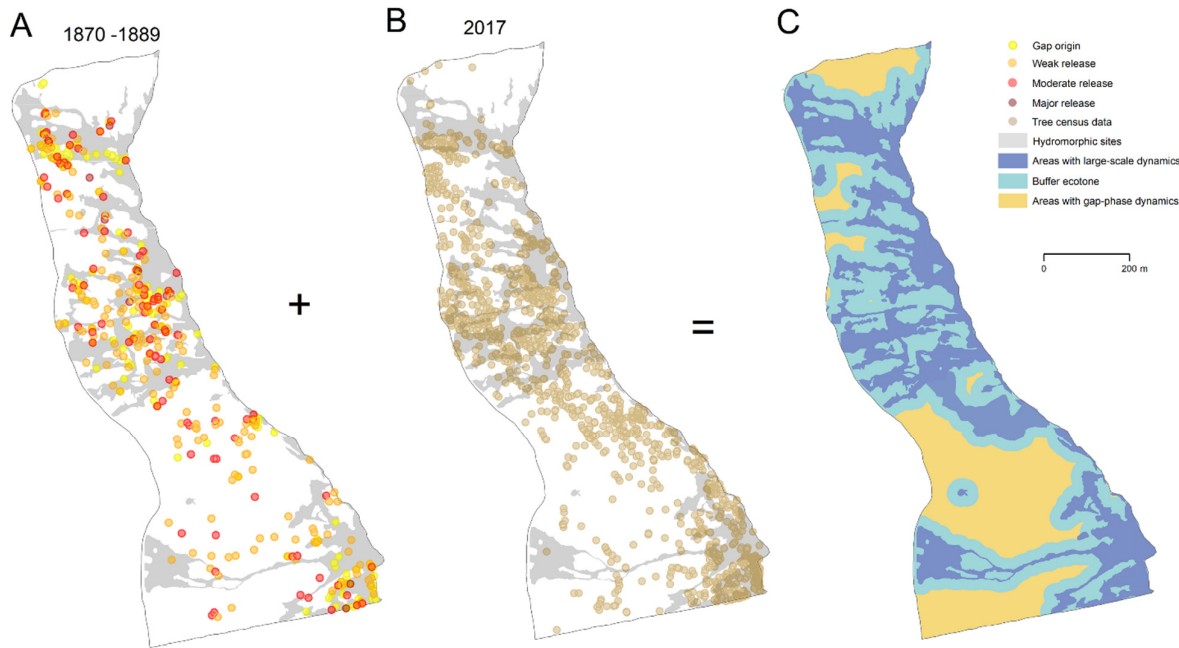

**Figure 4.** Maps of the spatial pattern of severe disturbances in Boubin, based on dendrochronological (**A**) and tree census (**B**) data and a map of the spatially-explicit mixed disturbance regime (**C**), derived on the basis of the soil hydromorphism map plus dendrochronological and tree census data.

The summary disturbance history of areas with terrestrial soils differed from hydromorphic sites in both the canopy area disturbed and gap origin (Figure 5). Although the mean disturbance rate was roughly the same, about 8% per decade for both datasets, trees on hydromorphic soils showed a lower reaction disturbance rate throughout most studied periods, with substantial canopy loss in the 1840s and 1870s (15.9% CAD, 27.7% CAD, respectively) after the series of windstorms and consequent outbreak of *Ips typographus* L. (Figure 5B, upper part). In contrast, on terrestrial soils we identified more or less stable canopy disturbances in every decade (Figure 5A, upper part), with two periods of higher canopy disturbances in the 1870s (20.5% CAD) and 1980s (24.6% CAD), due to the massive dye off of *Abies alba* enhanced by atmospheric pollution [66] as well as drought episodes [67] in Central Europe. While regeneration in gaps dominated hydromorphic stands throughout almost entire 19th century and culminating in the 1890s (Figure 5B, bottom part), trees occupying terrestrial stands poorly regenerated in gaps (Figure 5A, bottom part), with the exception of the period following windstorms in the 1870s.

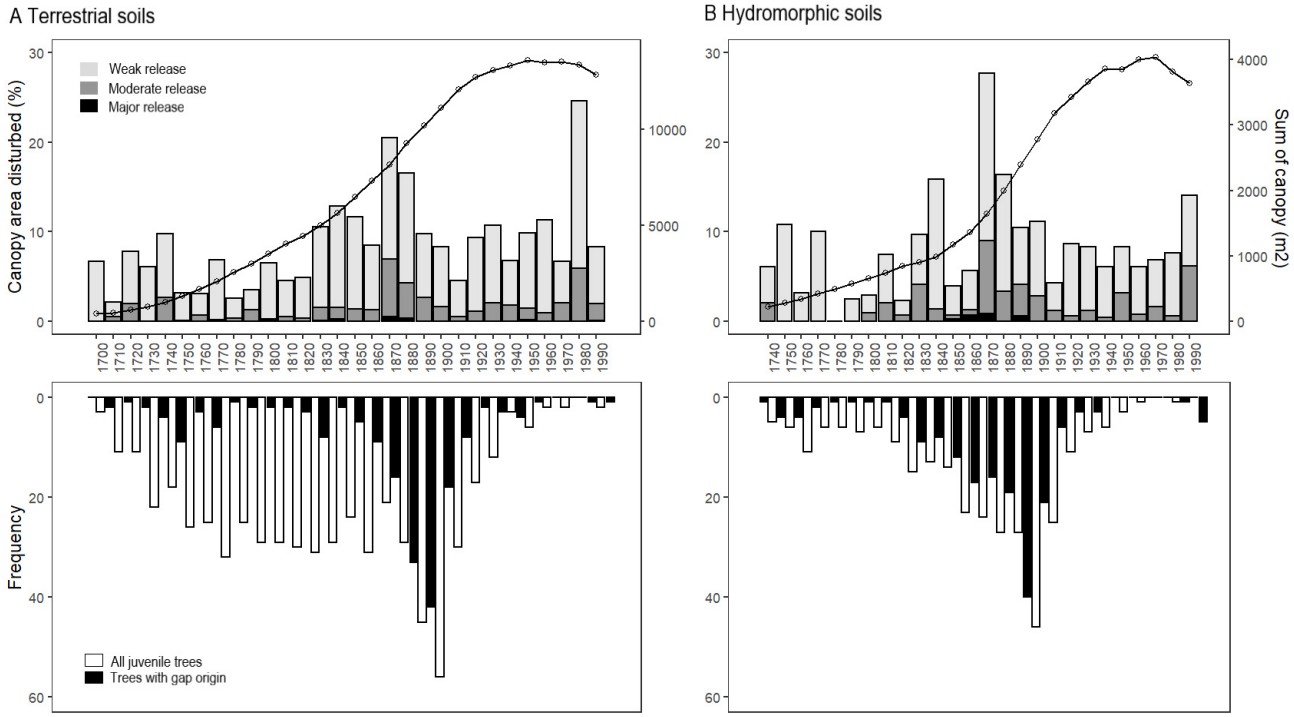

**Figure 5.** Summary disturbance histories of Boubin, developed separately for: (**A**) terrestrial and (**B**) hydromorphic soils. Upper charts represent the proportion of disturbed canopy area (%) from the total sum of exposed canopy (sample depth, m²) in the examined decade. Bottom charts depict the number of trees indicating gap origin.

Variograms showed distinct differences in spatial characteristics between hydromorphic and terrestrial soils (Table 3, Figure S1). Hydromorphic soils yielded slightly longer ranges (18.8 for weak release, 20.9 for moderate release) than terrestrial sites (11.0 for weak release, 8.0 for moderate release). While the exponential model was the most successful model for terrestrial soils, The Gaussian and stable models best fitted the hydromorphic data (Table 3). This indicates the concentration of disturbed trees in a coherent structure of wider canopy openings on wetter sites, while gap-phase mortality processes dominate on drier soils.

**Table 3.** Characteristics of spatial autocorrelation for the best-fitting models. Range represents the distance beyond which observations are not spatially autocorrelated, ecologically interpreted as the diameter of the canopy gap. The nugget is referred to small-scale variation at the beginning of a model as a result of site variability or measurement error. The sill express the maximal variance of data set that is not spatially autocorrelated.

| Soil Hydromorphism | Release Intensity | Model | AIC | Range (m) | Nugget | Sill |
|---|---|---|---|---|---|---|
| Hydromorphic | Weak | Stable | −34.754 | 18.783 | 0.046 | 0.239 |
| | Moderate and major | Gaussian | −62.749 | 20.871 | 0.032 | 0.117 |
| Terrestrial | Weak | Exponential | −44.706 | 11.005 | <0.001 | 0.229 |
| | Moderate and major | Exponential | −79.914 | 8.040 | <0.001 | 0.117 |

*3.2. Historical Hypothesis (H₂)*

Trees surviving the Herwart storm exhibited unequivocally different disturbance histories compared to disturbed trees in terms of both frequency and severity, explicitly verifying hypothesis H₂. On average, tree survivors showed 40% more releases by number per tree than trees killed in 2017 (Table 2, Figure 6B). Differences between pairs became more pronounced when evaluating only moderate and major releases. Tree survivors

showed 90% more releases per tree than trees killed by Herwart. Similarly, the total sum of the boundary line percentage, representing the degree of the growth reaction, was significantly higher by 50% for survived trees than for disturbed trees (Figure 6D). The paired analysis also showed that trees untouched by the Herwart storm created a much larger total sum of disturbed canopy area (mean disturbed = 12.3 m², mean survivors = 30.5 m²) (Figure 6F). The differences in summary exposed area by disturbances may be due to the repeated exposure of a cored tree or due to its release at a later age when the crown area was larger. On the contrary to releases in radial growth, trees killed by the Herwart storm had a higher proportion of trees with gap origin (Figure S2B). In general, disturbed trees experienced about 50% more cases indicating juvenile growth in gaps than survivors.

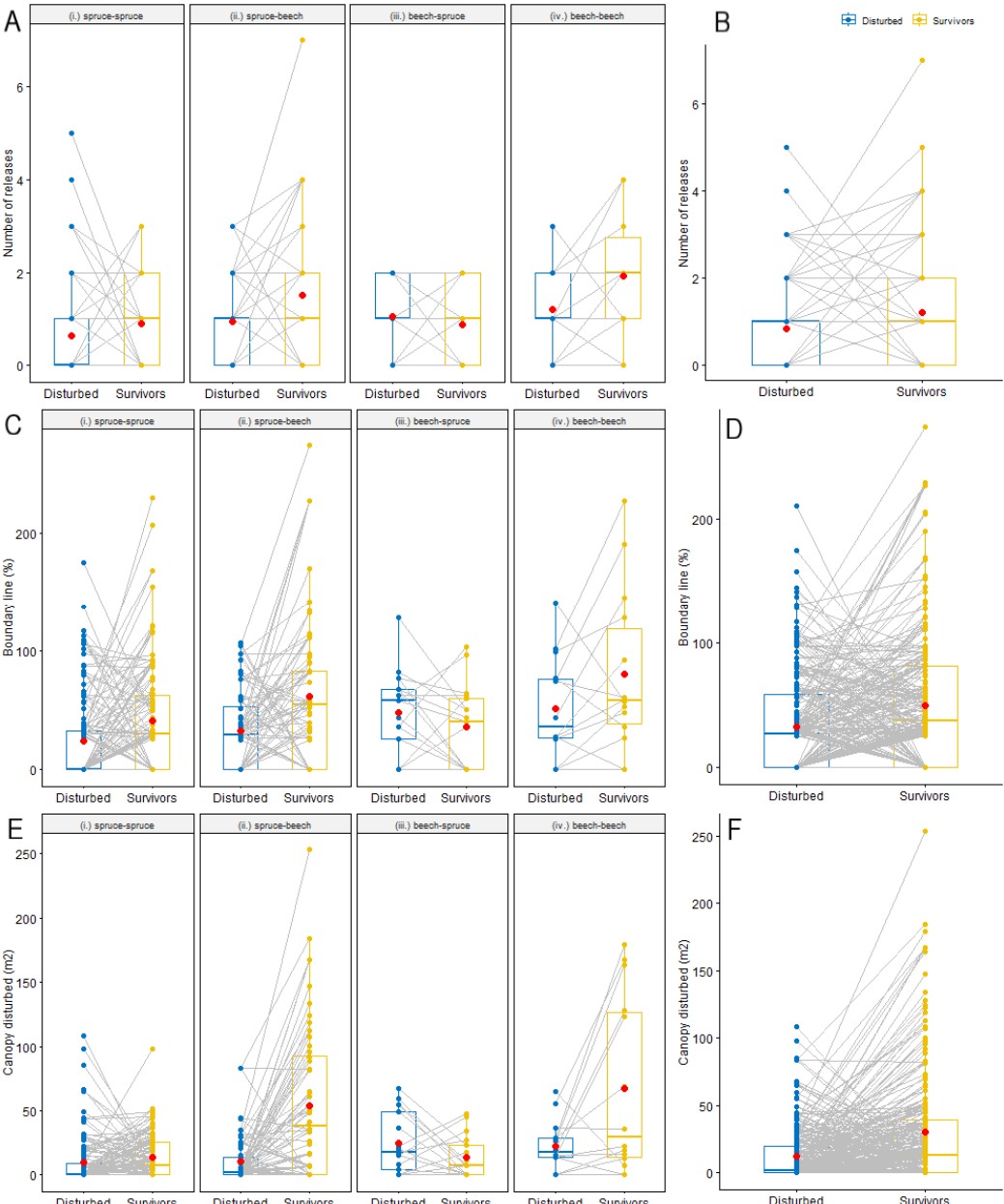

**Figure 6.** Paired boxplots of characteristics of disturbance frequency and intensity for both dendrochronological datasets for: (**A**) the number of releases sorted by species pairs; (**B**) the number of releases for all species; (**C**) the percent sum of boundary lines sorted by species pairs; (**D**) the percent sum of the boundary lines for all species; (**E**) the canopy area disturbed, sorted by species pairs; (**F**) the canopy area disturbed for all species. Blue and yellow boxes and points represent

disturbed and survivor datasets, respectively. The hinges correspond to the 25th and 75th percentiles, with the thick solid line showing the median and red circles the mean values. The upper and lower whisker ends represent 1.5 / IQR of the hinge, where IQR is the inter-quartile range. Points beyond the whisker ends are regarded as outliers. Grey lines connect individual trees within the pairs.

### 3.3. Species Hypothesis (H₃)

The species composition of trees killed by the storm significantly differed from the surviving community ($p < 0.001$, $\chi^2 = 45.25$), so we can reject the null hypothesis and confirm H₃ that the Herwart storm disproportionally affected some tree species populations. *P. abies* and *A. alba* were affected significantly more than expected by chance, while *F. sylvatica* was disturbed only in a minority of cases compared to its population abundance (Figure 7). Generally, spruce predominated in the set of disturbed trees (84%), followed by beech (11%) and fir (5%). On the contrary, spruce represented 54% of survived individuals, followed by beech (44%) and fir (2%). The most significant contributing tree species to the Chi-square score was *F. sylvatica* ($\chi^2 = 38.80$).

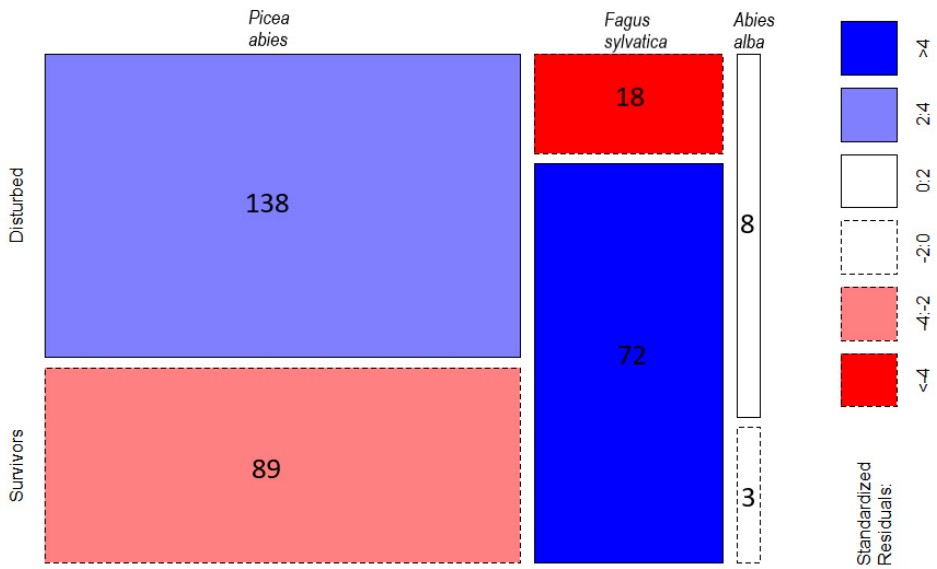

**Figure 7.** Mosaic plot of the frequencies of tree species distribution based on disturbed (observed) and survivor (expected) data sets. The area of cells is proportional to the frequency of elements in a contingency table. The color of the mosaic represents the relative magnitude of the value, based on standardized residuals, where blue indicates a higher value than expected and red indicates a lower value than expected.

On the other hand, most of the examined characteristics did not differ for most pairs of tree species, following a general trend (Table 2, Figure 6A,C,E), especially for the first two largest classes of tree species (i.e., spruce-spruce, spruce-beech). The only characteristic dependent on woody species was juvenile growth in a gap (Figure S2A), which emerged as a more important mechanism in the regeneration of spruce trees in comparison to beech trees that preferably recruited under a closed canopy.

### 3.4. Size Hypothesis (H₄)

According to our results of paired tests, we should reject our hypothesis that wind primarily selected larger and older individuals. Very surprisingly, trees disturbed by Herwart storm were characterized by significantly lower dimension and age ($p < 0.001$) in comparison to individuals surviving the disturbance (Table 2). On average, survivor trees reached nearly by 20% larger dimensions (mean survivors = 61.8 cm, mean disturbed = 52

cm). When comparing the tree-size structure, trees attacked by Herwart mostly occurred in the size category 30–70 cm, while in the dataset of survivor trees diameter classes 40–80 cm dominated (Figure S3A).

Age structure roughly outline the size structure (Figure S3C). On average, survived trees were older by 27 years than disturbed (mean disturbed = 201 yrs, mean survivors = 228 yrs). The youngest tree was 32 years old, while the oldest tree reached 567 years (*Abies alba*). Disturbed trees created a strong cohort of ages 120–180 years established after the windstorm period in the 1870s. As for intact trees, we distinguished two apparent cohorts: the first cohort of ages 120–140 years was dated after the same event as trees disturbed by Herwart; the second cohort was established 260–300 years before, indicating an origin after severe disturbance in the 1740s.

### 3.5. Material Hypothesis (H$_5$)

The paired test revealed a statistically significant difference in growth rates within pairs, seen in the mean tree-ring width over the whole life span of a tree ($p < 0.001$, Table 2). The most rapid growth rate was in trees killed by Herwart (mean = 0.93 RWI), while trees remaining intact after Herwart experienced slower increments (mean = 0.87 RWI). Together with results revealed within the Size hypothesis, these findings indicate a substantial divergence in growth strategies between two examined datasets. Life traits of slower growth facilitate trees to survive stronger windstorms via enhanced mechanical wood properties more than short-lived with fast growing strategy. They reach the canopy faster at the cost of earlier crown exposition to disturbances that does not permit trees to gain considerable dimension and age.

## 4. Discussion

### 4.1. Soil-Species Drivers of Wind Mortality Initiate Multiple Disturbance Regime

The mortality dynamics in studied old-growth forest was primarily driven edaphically through soil hydromorphism, thus determining species composition, disturbance regime and tree-growth rate. Severe wind disturbances shaped the forest structure non-randomly in both space and time, allowing the occurrence of several disturbance regimes covering different soil units. While gap-phase dynamics dominated terrestrial soils, hydromorphic sites rather experienced extremely strong, but less frequent events, such as Herwart or the series of windstorms in the 1870s. These results suggest the existence of two, edaphically-dependent, spatially-explicit disturbance regimes (Figure 4C). In contrast to mixed-severity disturbance regimes (e.g., [16,18,19]) highlighting the occurrence of several regimes mixed in time and space within a whole stand, our spatially-explicit mixed disturbance regime emphasizes the existence of spatially isolated, edaphically-determined regimes. Indeed, Figures 4C and S1 indicate that large-scale processes occupying hydromorphic areas are concentrated in relatively small, irregularly-shaped polygons as the difference in range of indicator variograms was not as dramatic as expected. Significant differences in many aspects of the disturbance regimes of nearby trees (up to 15 m away) suggest that quite sharp transitions between disturbance regimes are possible. This is in accordance with soil research, where sharp thresholds from terrestrial to fully hydromorphic soils have been found [37]. Based on soil evolutionary studies [68], we expect that the formation of spatial soil complexity is a comparatively slower process, exceeding the period of the forest cycle. Although individual trees may locally modify soil properties, the observed pedocomplexity was in principle established before the current generation of trees, and we thus might expect a certain invariability in the boundaries of disturbance regimes.

The existence of two spatially-explicit disturbance regimes under contrasting soil conditions well corresponds to and extend the studies of Daněk et al. [40] and Šamonil et al. [41]. The latter focused on the development of tree-populations on different soils within the Boubín forest. They found that while fine-scale mortality dominated drier terrestrial

soils, wetter hydromorphic sites were characterized by coarse-scale mortality following severe windstorms. Šamonil et al. [41] investigated how disturbances control pedodiversity of Central European beech-dominated primeval forest. They recorded fewer releases in trees occupying Gleysols and Histosols, suggesting unique disturbance regime of hydromorphic soils.

Between the two disturbance regimes described above, a buffer zone exists that merges the effects of both gap-phase and stand-replacing regimes (Figure 4C). While windthrow dynamics represent large-scale processes, bark beetle outbreaks participate in subsequent fine-scale processes. These sites are characteristically represented by terrestrial Albic Podzols in the immediate vicinity of water-affected areas [40]. It may be expected that this ecotone, characterized by substantial dynamics and flexibility, will be strongly affected by climate change accompanied by an abrupt transition in species composition. Due to the moisture limitations of Norway spruce in Central European forests [50] together with the effect of bark beetle outbreaks, such buffer zone will be the most suitable habitat for the ongoing expansion of European beech. This species is currently regenerating in large numbers under spruce canopies and has already come to dominate on terrestrial soils in Boubin. Along with the different mortality dynamics between the species, shifts in the disturbance regimes within this ecotone may be expected in the future and the boundaries of this zone will likely fluctuate depending on biotic disturbances.

As the proportion of uprooting among dead trees was more than 80% after Herwart storm [69], the higher tendency toward treefall described here is almost certainly related to the limited rooting stability of trees in soils with a high water table [34,35,70], decreasing resistive turning moment [71]. The edaphically-driven disturbance regime in Boubin therefore reflects the non-random spatial distribution of severe disturbance events as seen in Figure 4 and described by some authors for uprootings [3,36,72]. A higher incidence of uprooting on more saturated soils [73], accelerated by mild winter temperatures contributing to more frequent soil melting [46,74], would naturally lead to the pattern of repeatedly disturbed soils primarily on waterlogged areas as seen here.

The existence of two disturbance regimes, one of which is fundamentally controlled by strong disturbances and windthrow dynamics, will be necessarily reflected in the spatial pattern of hillslope processes. Phillips and Šamonil [75] considered the Šumava mountain forests, including Boubin, to have a biogeomorphological dominance of hillslope processes (see the state and transition model [76]). Regarding this concept, our study suggests that there may be different stages of such biogeomorphic succession within a forested stand and landscape. Parts of a landscape may be dominated by the mechanical influence of individual trees, but elsewhere, for example, a bark beetle outbreak can lead to a transition to an earlier stage of succession.

The distribution of tree populations along a gradient of soil moisture [40] resulted in a selective impact of the Herwart storm on conifers, primarily on Norway spruce. This is in line with Šamonil et al. [69], who found mortality rates after severe storms in a Central European old-growth forest of around 18–28% for *Picea abies* and 6% for *Fagus sylvatica*. In general, broadleaf species have been regarded as less vulnerable than conifers [23,77,78], especially during winter events when deciduous trees are defoliated, thus increasing their mechanical stability to wind [79]. Together with its deeper root system, this makes European beech more windfirm than silver fir [22] and especially Norway spruce [80], which is at highest risk of wind mortality due to very shallow rooting [26]. Despite the selective species-specific impact of the Herwart storm, we did not find any difference in species sensitivity for most characteristics studied that might determine a tendency to treefall secondarily through growth rate. Thus, our findings may be relevant to wide range of mountain forests across the temperate zone. The only significant species-specific characteristic was juvenile growth. While the dominant mode of rejuvenation of disturbed Norway spruces was gap origin, surviving European beeches rather recruited under the canopy. This observed difference in regeneration niches is in accordance with other studies from temperate and boreal old-growth forests of Europe [20,81,82].

### 4.2. Disturbed Trees Experienced Different Disturbance History Than Survivors

We also found that trees that survived the Herwart storm had a thoroughly different picture of forest disturbance history than those killed. These fragmental characteristics collect two mosaics of disturbance regimes. One is associated with the proportion of the forest killed by the storm. These trees usually recruited in gaps and characteristically experienced only one severe canopy accession event. On the other hand, trees surviving the Herwart storm characteristically germinated under the canopy and experienced several periods of suppression and release. We propose that this growth behavior in different canopy positions might be a key trait in driving the resistance of trees to failure, explaining why individuals with a more various disturbance history were more likely to survive the Herwart storm. Regarding the fact that periods of suppression preceding a release are distinguished by the production of wood of higher density for some species [83,84], longer times spent in the understory may give trees better mechanical support than those germinating in canopy openings and released once per life. Dependent on tree species, suppressed trees compensate for height growth limitations with lateral crown extension [85], which may in turn result in better stability during subsequent growth.

### 4.3. Preceding Growth Rate Determines the Tendency of Trees toward Fall

While our study supports the hypothesis $H_5$ that individuals with accelerated growth rates were more vulnerable to the Herwart storm, it does not support the size hypothesis $H_4$ that windstorms primarily select larger and older trees. Instead, trees killed by the Herwart storm had lower age and dimension than surviving neighbours, as opposed to most studies. Larger trees (and hence taller) are expected to be more predisposed to treefall than smaller [21,22,78,86–89] due to a higher tendency of trunk decay or higher wind forces at the top of the crown [21,23,90]. Nevertheless, several researchers have published contrasting findings. The review of Everham and Brokaw [78] proposed a unimodal or even bimodal relationship between tree size and the probability of treefall after catastrophic winds in many ecosystems. Peterson and Pickett [91] reported that the highest percentage of treefalls occurred in intermediate size classes. Some studies have suggested that wind damage increases to some maximum tree size and then ceases [92,93]. Others have even described a negative relationship between size and mortality after a hurricane in a tropical rainforest [94] or no relationship [95]. It is necessary to point out that this survivor data set does not represent the overall DBH distribution in Boubin that has a characteristic left skewed distribution (Figure S3B). We might hypothetically claim that Herwart selectively removed the biggest trees in the forest. However, this is not a representative data set for comparison, since it contains a considerable proportion of juvenile trees that might be suppressed. Instead, we believe comparing the two datasets including predominantly exposed trees is more appropriate. In addition, our results support the unimodal relationship hypothesis [78] that the smallest and largest trees are better adapted to resist the most extreme windstorms. According to this model, the greatest damage is found in intermediate stem size classes with a higher proportion of uprooting, while larger stems characterized by snapping have a lower percentage of treefall risk. Likewise, shedding branches could be a contributing adaptive strategy to reduce the area of trunks for wind to push on, thus allowing large trees to resist even high wind loads [21,35].

Tree age as a determinant of disturbance susceptibility has not been sufficiently examined [23] and existing results are rather confounding. Whereas some researchers have found treefall probability increasing with age in Central European mountain forests and US hardwood temperate forests [77,96,97], in a southern US boreal forest determined higher mortality rates in mature than in older stands [23]. They attributed this to the fact that older stands are dominated by late-successional, shade-tolerant species that are generally more windfirm [23,77,78,98]. We found a similar trend, with disturbed trees distinguished by shorter lifespans. The most probable explanation why younger and smaller

individuals are more predisposed may be that rot is a key factor limiting trees at water-saturated areas. A root system weakened by pathogens might be more prone to wind disturbance. Finally, frequent and severe windstorms may not allow trees to grow to exceptional age [32].

Different tree-growth behavior is apparently a key factor determining the survival of stems in our study area. As mentioned above, individuals with accelerated growth rates were more vulnerable to the Herwart storm than those slower growing during their life span. This is in line with Meyer et al. [95], who found higher mean growth increments for spruce trees damaged by windstorm Lothar in 1999 in Switzerland compared to undamaged trees. It is commonly accepted that smaller growth rates result in wood of increased density, providing a mechanical advantage in wind firmness (e.g., [99,100]). Indeed, the relationship between tree-ring width and wood density differs among species. While wider spruce tree-rings were found to reduce wood density in most studies, results for beech are contradictory (e.g., [61,101]). Thus, our results on increased probability of wind mortality of trees growing faster might be explicated better by the concept "grow fast, die young", concerning trade-offs between growth rate and tree lifespan [30,102,103]. According to this concept, trees growing more rapidly during early stages are associated with an accelerated life cycle. They reach the canopy faster, where they are exposed to disturbance, or may not effectively invest in disease protection. Undoubtedly, edaphic conditions beneath trees can contribute to accelerated growth rates, especially on soils partly saturated by water (e.g., Stagnosols). Assuming increased moisture-limited growth for some species on drier sites in Boubin [50], trees on hydromorphic sites might benefit from the higher water table and invest in larger increments, while in turn being more prone to wind disturbances in the future.

### 4.4. Results in a Broad Context and Implications for Forest Management

Boubin forest is a typical representative of mixed mountain old-growth forest, serving as an "etalon" in the formation of traditional concepts of close-to-nature forest management [104,105]. Thus, our results might be extrapolated to analogical natural forests across the Europe, e.g., Novohradske Mts. in Central Europe [16], the Carphatians [18,19,106], the Alps [11,107], the Balkan peninsula [22,108] and Apennines [109]. This study determined several mutually overlapping factors that might contribute to whole-stand resistance to severe storms. Unlike the moisture conditions, applied management strategies might rule further species, age and size structure of managed forests. Our findings may help forest managers predict and reduce storm damage through the implementation of natural processes into existing silvicultural systems. Despite limitations imposed by edaphic site conditions, this might be achieved by heterogeneous horizontal and vertical uneven-aged structures and diverse species composition. The Dauerwald concept [110], with the focus on retaining suppressed and older trees, is a promising approach. Still, it is clear that even old-growth forests are subject to infrequent severe disturbance events, extensively disrupting the forest structure.

## 5. Conclusions

This study yields insights into the key agents of mortality dynamics following severe windstorms in a temperate old-growth mountain forest. The results demonstrate that the susceptibility of trees to treefall is primarily driven edaphically, promoting the existence of a spatially-explicit, edaphically-determined disturbance regime. The distribution of tree populations along a gradient of soil moisture resulted in a selective impact of the Herwart storm, primarily affecting Norway spruce, indicating a tight soil-species-disturbance interaction. On the other hand, we did not detect any interspecific variation for most of studied characteristics. Together with ongoing changes in the species composition of the Boubin forest in favor of *Fagus sylvatica*, a shift in the disturbance regime is expected in the future.

The non-random spatial distribution of treefall events, largely limited to previously disturbed sites, indicates the extraordinary significance of windthrow dynamics in biogeomorphic processes. The combination of disturbance and hillslope processes with many feedbacks are likely important agents in the dynamics of primeval mountain forests, significantly increasing the forest structure complexity.

Our study supports the hypothesis that tree stability is related to previous tree growth and disturbance history in the vicinity. The functional trait of slow initial growth, with several periods of suppression, provides a tree mechanical advantage in wind firmness, as well as delayed canopy accession. In contrast, trees with rapid growth rates are connected with accelerated life cycle, and thus, earlier predisposition to root pathogens as well as exposure to disturbance. The presence of such ancient trees with high-density wood might thus increase the whole-stand resistance to severe storms. Our results emphasize that local conservation management strategies for forested landscape should integrate natural processes into existing silvicultural systems to reduce storm damage on the landscape-level.

**Supplementary Materials:** The following are available online at www.mdpi.com/article/10.3390/f12111599/s1, Methods S1: A detailed description of boundary line calculation, Methods S2: A detailed description of spatial autocorrelation calculation, Figure S1: Indicator variograms of severe disturbances in the 1870–1899 period, Figure S2: Mosaic diagrams of trees regenerating in gaps or under the canopy, Figure S3: Histograms of DBH and tree age of disturbed and survivor trees.

**Author Contributions:** Conceptualization, I.V. and P.Š.; methodology, I.V. and P.Š.; software, I.V., J.K., A.R.-S., T.C. and D.A.; validation, I.V., P.Š., J.K., A.R.-S. and T.C.; formal analysis, I.V., P.Š., J.K., A.R.-S. and T.C; investigation, I.V., P.Š., J.K., A.R.-S. and T.C; resources, I.V.; data curation, I.V., P.Š., J.K. and D.A.; writing,—Original Draft Preparation, I.V. and P.Š.; writing—Review and Editing, I.V.; visualization, I.V., A.R.-S. and D.A.; supervision, P.Š.; project administration, P.Š.; funding acquisition, P.Š. All authors have read and agreed to the published version of the manuscript.

**Funding:** This research was funded by Czech Science Foundation, project no. 19-09427S, and Technology Agency of the Czech Republic, project no. SS02030018.

**Data Availability Statement:** Data are available on request from the corresponding author.

**Acknowledgments:** We are grateful for permission given for our work by the Administration of the Šumava National Park and Protected Area. We thankfully acknowledge all the colleagues of Blue Cat team responsible for field surveys and sample processing, especially Daniel Cigánek. Our special thanks go to David Hardekopf for English proofreading and reviewers for their constructive comments and suggestions that considerably improved the paper.

**Conflicts of Interest:** The authors declare no conflict of interest.

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
