# Peer review of "Dead or Alive: Drivers of Wind Mortality Initiate Multiple Disturbance Regime in a Temperate Primeval Mountain Forest"

_forests, doi:10.3390/f12111599_

Round 1

Reviewer 1 Report

  1. A fundamental question on the method of determining the calendar age of trees. Section 2.3 “Sampling design and data processing” describes a method for coring at a height of 0.5-1.3 m (line 168). How did the authors determine the calendar age of a tree using such increment cores? Typically, cores taken from the base of the tree trunk are used. Taking a core at a height of 0.5-1.3 m reduces the age of the tree by 10-30 years.
  2. To analyze juvenile growth, the central portion of the core less than 3 cm from the pith was analyzed (line 198-199). However, the analysis of the spectrum of annual increments taken at a height of 0.5-1.3 m does not reflect the juvenile period of the tree's life. Especially for spruces with slow growth in the first 5-7 years, they can even reach a height of 1.3 m in a window by 15-20 years. How did the authors solve this question methodically?
  3. Table 2 (line 251). The heading requires a meaningful expansion. In the table, insert a column indicating the sample size. In the note, it is advisable to decipher all the abbreviations used in the first column. Many of the table rows are not centered in height.
  4. There is a repeat of the text of line 274-275 and lines 198-199.
  5. In Figure 5 (line 349), there is no meaningful explanation of the line at the top of the graph (sum of canopy).
  6. It is necessary to expand the title of the table 3 (line 362).
  7. In Figure 7 (line 398), in the upper field, I recommend giving the full name of the tree species instead of abbreviations.

Reviewer 2 Report

Dear authors, many thanks for the interesting and relevant research paper!

The overall quality of the reviewed article is good, the presentation is understandable, and the text is easily comprehendible. However, minor spellcheck is required to eliminate the infrequent misprints and other errors. 

The research makes use of a great dataset obtained using modern approaches, techniques, and tools.

Regarding the clarity of the presentation, sections 3.4 and 3.5 require better elaboration to be clearer for the reader.

The research, as some of the cited papers, focuses on the Boubin forest in the Czech Republic. My concern is about the potential for reusing the data presented in the paper. Extrapolation and application beyond the research area are better to be outlined within Introduction and elaborated in the Discussion sections, as well as the prospects for further practical use of the research results by forest managers.

These, however, are minor points that could improve the overall value of the paper for a wide variety of potential readers. Readers with a scientific background are likely to find the paper interesting even in its present form.
